

# Reliability of motion phase identification for long-track speed skating using inertial measurement units

Tomoki Iizuka[1,2] and Yosuke Tomita[2]

[1] Department of Rehabilitation, Kurosawa Hospital, Takasaki, Gunma, Japan
[2] Department of Physical Therapy, Graduate School of Health Care, Takasaki University of Health and Welfare, Takasaki, Gunma, Japan

## ABSTRACT

**Background**. Precise identification of motion phases in long-track speed skating is critical to characterize and optimize performance. This study aimed to estimate the intra- and inter-rater reliability of movement phase identification using inertial measurement units (IMUs) in long-track speed skating.

**Methods**. We analyzed 15 skaters using IMUs attached to specific body locations during a 500m skate, focusing on the stance phase, and identifying three movement events: Onset, Edge-flip, and Push-off. Reliability was assessed using intraclass correlation coefficients (ICC) and Bland-Altman analysis.

**Results**. Results showed high intra- and inter-rater reliability (ICC [1,1]: 0.86 to 0.99; ICC [2,1]: 0.81 to 0.99) across all events. Absolute error ranged from 0.56 to 6.15 ms and from 0.92 to 26.29 ms for intra- and inter-rater reliability, respectively. Minimally detectable change (MDC) ranged from 17.56 to 62.22 ms and from 33.23 to 131.25 ms for intra- and inter-rater reliability, respectively.

**Discussion**. Despite some additive and proportional errors, the overall error range was within acceptable limits, indicating negligible systematic errors. The measurement error range was small, demonstrating the accuracy of IMUs. IMUs demonstrate high reliability in movement phase identification during speed skating, endorsing their application in sports science for enhanced kinematic studies and training.

## INTRODUCTION

Speed skating is a competitive sport where athletes strive to achieve the fastest time over a set distance on ice. In long-track speed skating, competitors navigate a 400-meter oval rink, often exceeding speeds of 50 km/h. Long-track speed skating demands a mastery of gliding technique and kinematic characteristics, both crucial to performance optimization (*Konings et al., 2015*). Previous research has highlighted the significance of various factors such as knee joint and trunk angles (*Noordhof et al., 2013*; *Noordhof et al., 2014*; *Van Ingen Schenau, De Groot & De Boer, 1985*), and the push-off angle, which is the tibia's frontal plane angle relative to the ice, in enhancing power output and skating velocity.

Long-track speed skating, similar to walking and running, necessitates the precise identification of movement phases using valid and reliable methods to effectively analyze

Corresponding author
Yosuke Tomita,
tomita-y@takasaki-u.ac.jp

kinematic characteristics. While a universally accepted classification for these movement phases in speed skating is lacking, the instances of blade contact with and departure from the ice are typically considered as the onset and offset of the stance phase, respectively. In speed skating, appropriate transfer of the edge during the stance phase is important. The relationship between floor reaction forces in the stance phase and performance has also been documented in previous studies (*Yuuki, Ae & Asami, 1992*). This indicates that the stance phase is important for generating speed in speed skating (*Yuuki, Michiyoshi & Fujii, 1996*). A skating blade is approximately one mm wide and has two edges: the outer and inner edges. The outer and inner edges of a speed skating blade refer to the sides of the blade that face away from and toward the skater's body, respectively. In the straight segment, the outer edge first contacts the ice (Onset: ON), while the blade rolls over to the inner edge during the middle of the stance phase. This transfer of the edge from outer to inner is referred to as Edge-flip (EF) in this study. After EF, push-off (OFF) is characterized by a sudden increase in the angular velocity of knee extension (*Yuda et al., 2007*).

The kinematic characteristics of speed skating have predominantly been studied using video camera analysis (*Yuda et al., 2007*). However, this method faces several challenges, including obstructions to visibility by objects or individuals on the ice, necessitating multiple cameras for optimal coverage. Moreover, video analysis is time-consuming and labor-intensive, particularly in the calculation of joint angles and movement events, due to the need for extensive landmark coordinate analysis. Consequently, while video cameras offer a non-invasive analytical approach, their limitations hinder their effectiveness in speed skating training.

In recent years, the use of inertial measurement units (IMUs) has emerged as a solution to the constraints of video camera analysis in sports science. IMUs, comprising a tri-axial accelerometer, gyroscope, and magnetometer, facilitate the real-time calculation of acceleration and angular velocity (*Faisal et al., 2019*). This technology has been applied in various sports, including running (*Zrenner et al., 2020*), skiing (*Bessone, Petrat & Schwirtz, 2019*; *Meyer et al., 2022*), and short-track speed skating (*Kim et al., 2019*), for kinematic analysis. The use of IMUs in sports science provides several advantages. First, IMUs are less influenced by the measurement environment compared to video cameras. As kinematic measurement with IMUs does not require video cameras, it is not restricted by the problems of missing markers and limited measurement range. Secondly, the acceleration and angular velocity are measured in real-time, allowing for the immediate calculation of joint angles and events during motion. Thus, the use of IMUs in sports science provides a promising alternative to video camera analysis, allowing for more precise and comprehensive data collection in real-time.

A few studies utilizing IMUs in long-track speed skating have also been reported (*Van der Kruk et al., 2018*), but the application of IMUs in long-track speed skating has been limited to date. Furthermore, the validity and reliability of the kinematic analysis using IMUs have not been well established. In a previous study, we reported that movement event identification using IMUs was comparable with that using a foot pressure system, where the relative error ranged within 3.6% (*Tomita et al., 2021*). However, the reliability of the analysis method has not been established in identifying detailed movement events

**Table 1 The demographics of the study participants ($n = 15$).**

|  | Male ($n = 5$) | Female ($n = 10$) |
| --- | --- | --- |
| Age, years (mean ±SD) | 16.80 ± 1.47 | 19.20 ± 1.94 |
| Personal Best Time for 500m, s (mean ±SD) | 37.67 ± 1.62 | 41.38 ± 1.90 |
| Measured Time for 500m, s (mean ±SD) | 38.39 ± 1.62 | 42.00 ± 1.68 |

**Notes.**

Measured time for 500m: the 500m time for the experimental race.

**Table 2 Sensor locations.**

| | |
| --- | --- |
| Lower Thoracic | In line with the spinal column at L1/T12. In the direction of the vertical line. |
| Pelvis | Middle of the sacrum. In the direction of the vertical line. |
| Thigh | Half on the line from the anterior spina iliaca superior to the superior part of the patella. In the direction of the line from the anterior spina iliaca superior to the superior part of the patella. |
| Shank | One-third on the line between the tip of the fibula and the tip of the medial malleolus. In the direction of the line between the tip of the fibula and half-center between the tip of the medial malleolus and lateral malleolus. |

including EF and PO. Therefore, the objective of this study was to estimate the intra- and inter-rater reliability of movement phase identification using IMUs in long-track speed skating. We employed intraclass correlation coefficients and Bland-Altman analysis to evaluate overall reliability and the presence of systematic error.

## MATERIALS & METHODS

### Participants

Fifteen long-track speed skaters participated in this study. We included participants with a relatively large age range, from middle school to university level, to increase the generalizability of the study (Table 1). Written informed consent was obtained from all participants, as well as from the guardians of participants aged under 18 years. The study was approved by the Ethics Committee of Takasaki University of Health and Welfare (No. 1904).

### Data acquisition

Kinematic data were collected at the Meiji Hokkaido Tokachi Oval, an indoor 400 m rink. The data were obtained using six IMU sensors (myoMOTION, Noraxon, Scottsdale, AZ, USA) at a sampling rate of 200 Hz. The acquired kinematic data were filtered with a Butterworth low-pass filter (6th order, cut-off frequency: 20 Hz). The sensors were attached to the lumbar spine, pelvis, bilateral thighs, and bilateral lower legs according to the manufacturer's specifications (Fig. 1, Table 2). Participants completed a full-speed 500 m skate on double track from a stationary position for the measurement, where the length was approximately 110 m and 100 m for the straight and curve sections, respectively.

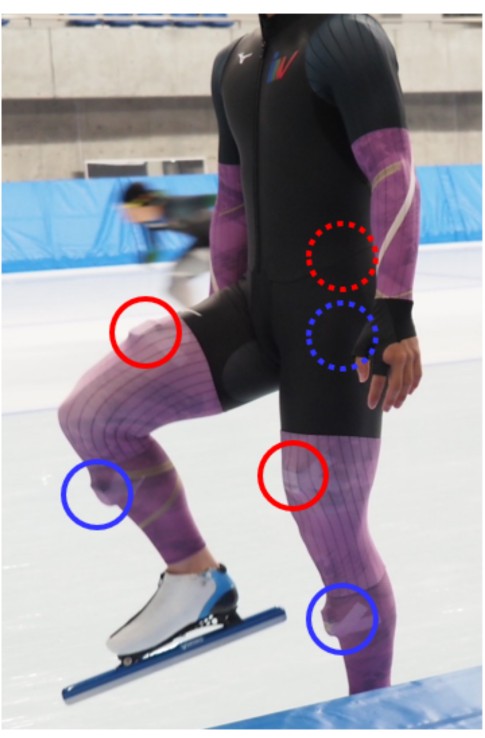

**Figure 1** **IMU sensors attached to a speed skater.** Red dotted circle, lower thoracic. Blue dotted circle, Pelvis. Red circle, Thigh. Blue circle, Shank. UT and Pel are affixed to the back and cannot be seen directly.

## Data analysis

In our analysis, we excluded data from the acceleration section from the start to 100 m due to significant differences in skating technique. Thus, the analysis included a 400 m section (straight: approximately 110 m × 2, curve: approximately 100 m × 2) from 100 m to 500 m. We first identified each stroke based on a consistent pattern of knee flexion angles (*Tomita et al., 2021*), as shown in Fig. 2. In the present study, each stroke was defined as the period from one push-off to the subsequent ipsilateral push-off. The beginning of each stroke was defined as the time when the knee flexion angle reached its minimum value, which corresponds to the beginning of the swing phase (vertical black lines in Figs. 2, 3 and 4). The timing of ON, EF, and PO after the beginning of each stroke was separately calculated (Fig. 2). EF was calculated only for the straight. ON and PO were calculated for the left lower limb at the curve using the same criteria as for the straight, while the right lower limb was calculated using different criteria because of the significant difference in joint motion patterns (see Supplementary materials). The timing data were compiled during each skating section (straight and curve for each stroke side (right and left)).

## Onset

The onset (ON) event identification was referenced to the knee flexion angle and the vertical acceleration of the Shank sensor (red dotted lines (A), Fig. 3, Table 3). For straight right and left and curve left, the timing of the minimum impact of the first shank anterior-posterior

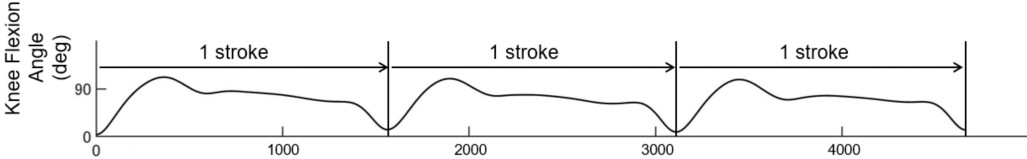

**Figure 2  1 stroke defined with respect to the minimum peak of the knee joint angle.** The knee flexion angle is defined as the relative angle between the thigh and shank segments in the sagittal plane.

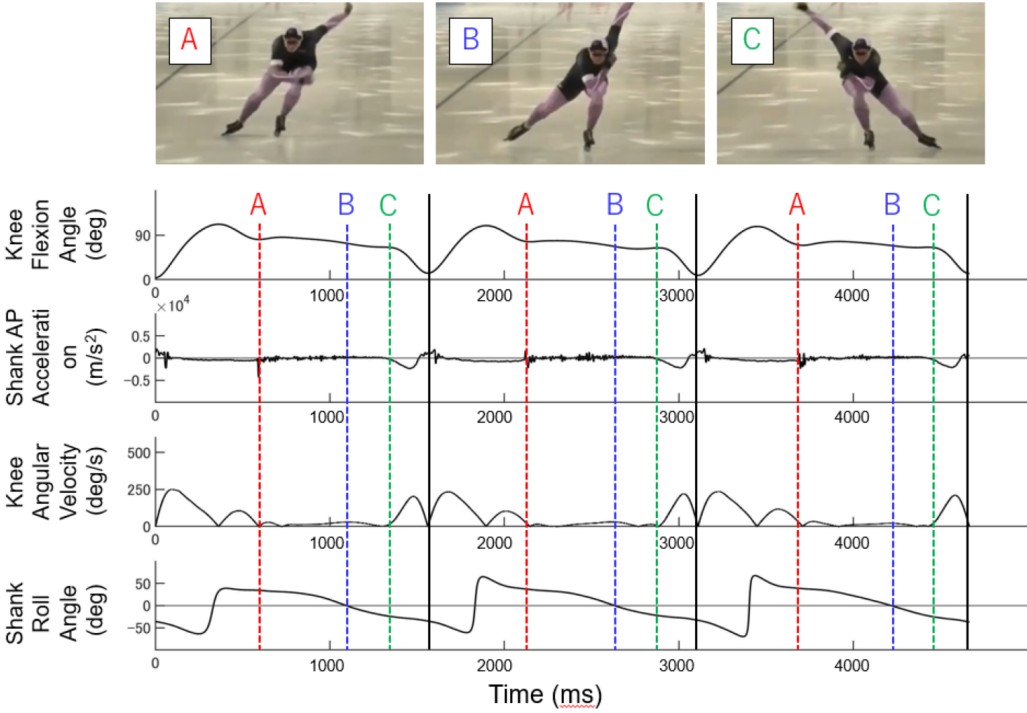

**Figure 3  Identification of each event in bilateral straight and left lower limb curve.** Knee flexion angle, the relative angle between the thigh and shank segments in the sagittal plane; Shank AP Acceleration, Vertical acceleration of Shank sensor; Knee Angular Velocity, Knee flexion angular velocity; Shank Roll Angle, Roll angle of shank sensor (the segment angle of the shank in the frontal plane); Black lines, 1 stroke cycle; Red dotted lines (A), Onset; Blue dotted lines (B), Edge-flip; Green dotted lines (C), Push-off.

(AP) acceleration after the maximum peak of the knee flexion angle was used as the reference. The shank AP acceleration was derived from the sensor attached 1/3 of the way on the line between the tip of the fibula and the tip of the medial malleolus (Table 2). On the curve right, the timing when the impact of the first shank-AP acceleration reached its minimum value after the first peak of the knee flexion angle was used as the reference (red dotted lines (A), Fig. 4).

## Edge-flip

The edge-flip (EF) is the translation of the blade from the outer- to the inner-edge that only occurs during the straight section. Therefore, EF was identified only in the straight section.

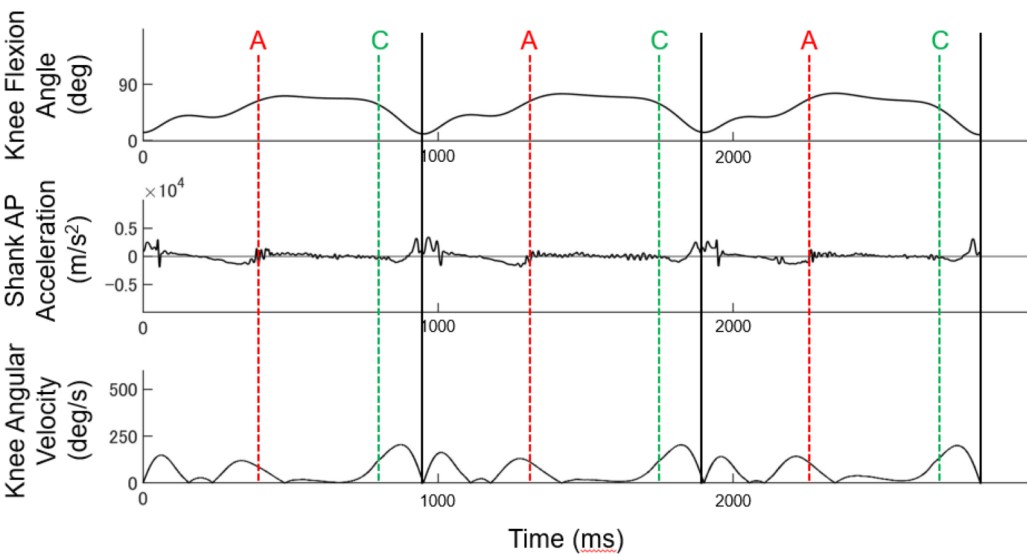

**Figure 4  Identification of each event in the right lower extremity curve.** Knee flexion angle, the relative angle between the thigh and shank segments in the sagittal plane; Shank AP Acceleration, Vertical acceleration of Shank sensor; Knee Angular Velocity, Knee flexion angular velocity; Black lines, 1 stroke cycle; Red dotted lines (A), Onset; Green dotted lines (C), Push-off; note that Edge-flip is not shown since Edge-flip is absent in the curve section.

The EF event identification was referenced to knee flexion angle, vertical acceleration, and roll angle of the Shank sensor. The EF was defined as the time at which the roll angle of the shank reached 0 deg after ON (blue dotted lines (B), Fig. 3, Table 3).

### Push-off

The push-off (PO) event identification was referenced to knee flexion angle, knee angular velocity, and shank-AP acceleration. For both straight and curve, the PO was identified by finding the time point at which the shank AP acceleration started to decrease and the knee angular velocity increased after the EF (green dotted lines (C), Figs. 3 and 4, Table 3).

The event identification of ON, EF, and PO was performed twice by each examiner (T.I. and Y.T) by visual identification according to the above-mentioned definitions.

## STATISTICAL ANALYSIS

### Calculation of Intraclass Correlation Coefficient

Intra-rater reliability was assessed using Intraclass Correlation Coefficient (ICC) (1,1), while inter-rater reliability was assessed using ICC (2,1). The 95% confidence intervals (95% CI) were calculated for each ICC. Statistical analysis was performed using SPSS ver. 22 (IBM, Armonk, NY, USA), and a significance level of 5% was used.

### Bland-Altman analysis

A Bland-Altman plot was created to examine the presence of systematic errors. This involved creating a scatter plot with the difference between the two measurements (d) on the $y$-axis and the mean of the two measurements on the $x$-axis (*Bland & Altman, 1986*).

**Table 3 Data and criteria referenced at each phases.**

|  |  | Data used | Criteria |
|---|---|---|---|
| Onset (ON) | Straight (Rt & Lt) Corner (Lt) | Knee flexion angle Shank-AP acceleration | Timing of the minimum impact of the first shank anterior-posterior acceleration after the maximum peak knee flexion angle. |
|  | Corner (Rt) | Knee flexion angle Shank-AP acceleration | Timing of the first shank anterior-posterior acceleration reaching its minimum after the first peak of knee flexion angle. |
| Edge-flip (EF) | Straight (Rt & Lt) | Shank roll angle | Timing when shank roll angle reaches 0 deg after ON. |
| Push-off (PO) | Straight (Rt & Lt) Corner (Rt & Lt) | Knee flexion angle Shank-AP acceleration Knee angular velocity | The timing at which the shank anterior-posterior acceleration begins to decrease and the knee angular velocity increases. |

**Notes.**
Rt, Right side; Lt, Left side; Shank AP Acceleration, Anterior-posterior acceleration of Shank sensor; Knee Angular Velocity, Knee flexion angular velocity; Shank Roll Angle, Roll angle of shank sensor.

We evaluated two types of systematic errors; additive and proportional errors. Additive error was evaluated by calculating the Limit of Agreement (LOA) using the 95% CI of the mean of the difference between the two measurements. We considered that additive error may be present when the 95% CI of the difference between the measurements did not cross zero. Proportional error was evaluated by calculating a regression coefficient (r) of the relationship between the two measurements (*Anvari, Halpern & Samir, 2018*). We considered that proportional error may be present when the correlation reached statistical significance.

## Calculation of SEM (standard error of measurement) and MDC (minimal detectable change)

$MDC_{95}$, a measure of absolute reliability, was calculated to examine chance error; $MDC_{95}$ was calculated using the following Eq. (1) (*Romijnders et al., 2021*):

$$MDC_{95} = SEM \times 1.96 \times \sqrt{2}. \tag{1}$$

The standard error of measurement (SEM) required to calculate $MDC_{95}$ was calculated by deriving the mean square error from the test and retest measurements by repeated one-way ANOVA and using the formula formula (2) below (*Romijnders et al., 2021*; *Hansen et al., 2022*; *Falbriard et al., 2018*):

$$SEM = \sqrt{MSE}. \tag{2}$$

The time from offset to offset on the same side was defined as stroke duration. The %SEM and %MDC$_{95}$ were calculated by normalizing SEM and MDC$_{95}$ using the stroke duration (Fig. 1).

## RESULTS

There were 15 subjects (5 males: 10 females, age: 18.4 years). The strokes analyzed totaled 164 straight strokes and 250 curve strokes for both the right and left.

### Intra-rater reliability

Intra-rater reliability results are shown in Table 4, with ICC(1,1) values ranging from 0.86 to 0.99 for all movement events. There was no consistent difference in ICC(1,1) values between the right and left sides in the straight, but the right side tended to have lower ICC values in the curve. The %SEM ranged from 1 to 3% for all items. The %MDC$_{95}$ ranged from 2 to 7% for all movement events.

The Bland-Altman plot for intra-rater reliability is shown separately for each movement event (Fig. 5). The Bland-Altman plot analysis showed additive errors in the straight ON for the left and right lower limbs, EF for the left and right, and PO for the left and right lower limbs and in the curve ON for the left and right, and PO for the right lower limbs. Proportional errors were also observed for the ON for the left lower limb in the curves (Table 4).

### Inter-rater reliability

Inter-rater reliability results are shown in Table 5, where ICC(2,1) values ranged from 0.81 to 0.99 for all movement events. There were no consistent side-specific differences in ICC(2,1) values on the straight, but the right side tended to have lower ICC values in the curve. The %SEM ranged from 1 to 5% for all events. The %MDC$_{95}$ ranged from 3 to 15% for all movement events.

Bland-Altman plots of inter-rater reliability are shown separately for each movement event (Fig. 6). Additive errors were observed in the straight ON for the left, EF for the right, and PO for the right and left lower limbs, and in the curve ON for the left, and PO for the left lower limb. Proportional errors were also observed in the straight ON right lower limb, EF for the right lower limb, and PO for left and right lower limbs, and the curve ON right lower limb and PO for left and right lower limbs (Table 5).

## DISCUSSION

The objective of this study was to estimate the reliability of identifying motion phases during long-track speed skating runs and to investigate systematic and measurement errors.

The results of this study indicated high intra-rater reliability, with ICC(1,1) values greater than 0.8 for all items. Our quantitative analysis showed that proportional errors may be present only for the ON for the left lower limb in the curve. Furthermore, several items exhibited additive errors with Limits of Agreement (LOAs) not containing zero, while the positive and negative values were not consistent. Nevertheless, the range of any additive

Iizuka et al. (2024), *PeerJ*, DOI 10.7717/peerj.18102

**Table 4  ICC for intra-rater reliability.**

| | | Stroke duration, ms (SD) | d, ms | LOA, ms upper; lower limits | ICC (1,1) | 95%CI upper; lower limits | SEM, ms | %SEM | MDC$_{95}$, ms | %MDC$_{95}$ | r (p) |
|---|---|---|---|---|---|---|---|---|---|---|---|
| **Straight** | | | | | | | | | | | |
| ON | Rt | 1,236.22 (116.84) | −3.11 (0.59) | −4.28; −1.95 | 0.86 | 0.82; 0.89 | 15.79 | 0.01 | 43.76 | 0.04 | 0.06 (0.25) |
| | Lt | 1,211.97 (120.67) | −3.25 (0.28) | −3.80; −2.70 | 0.97 | 0.91; 0.98 | 8.49 | 0.01 | 23.54 | 0.02 | 0.06 (0.30) |
| EF | Rt | – | −0.56 (0.51) | −1.57; 0.44 | 0.98 | 0.98; 0.99 | 9.87 | 0.01 | 27.37 | 0.02 | 0.09 (0.09) |
| | Lt | – | −0.66 (0.41) | −1.46; −0.14 | 0.99 | 0.99; 0.99 | 8.49 | 0.01 | 28.82 | 0.02 | 0.01 (0.98) |
| PO | Rt | – | 1.63 (0.62) | 0.41; 2.85 | 0.98 | 0.97; 0.98 | 16.03 | 0.01 | 44.43 | 0.04 | 0.09 (0.11) |
| | Lt | – | 1.96 (0.71) | 0.56; 3.35 | 0.97 | 0.97; 0.98 | 18.37 | 0.02 | 50.92 | 0.04 | 0.07 (0.23) |
| **Curve** | | | | | | | | | | | |
| ON | Rt | 885.80(107.42) | −2.52 (0.40) | −3.30; −1.73 | 0.91 | 0.88; 0.93 | 13.15 | 0.02 | 36.44 | 0.04 | 0.01 (0.83) |
| | Lt | 888.24(108.30) | −3.07 (0.26) | −3.57; −2.56 | 0.95 | 0.90; 0.97 | 6.33 | 0.01 | 17.56 | 0.02 | 0.16 (<0.01) |
| PO | Rt | – | 6.15 (0.66) | 4.86; 7.43 | 0.95 | 0.92; 0.97 | 22.45 | 0.03 | 62.22 | 0.07 | 0.03 (0.48) |
| | Lt | – | 0.75 (0.51) | −0.26; 1.75 | 0.97 | 0.97; 0.98 | 16.16 | 0.02 | 44.79 | 0.05 | 0.03 (0.58) |

**Notes.**

Note that stroke duration is presented only for ON since it is the intervals between each stroke.

SD, Standard deviation; ON, Onset; EF, Edge-flip; PO, Push-off; Rt, Right side; Lt, Left side; d, Difference between the two measurements; LOA, Limit of Agreement; ICC, Intraclass correlation coefficients; SEM, Standard error of measurement; MDC$_{95}$, Minimal detectable change; %SEM, SEM/stroke; %MDC$_{95}$, MDC$_{95}$/stroke; r (p), correlation coefficient ( *p*-value).

Iizuka et al. (2024), *PeerJ*, DOI 10.7717/peerj.18102

**Table 5  ICC for inter-rater reliability.**

| | | Stroke duration, ms (SD) | d, ms | LOA, ms upper; lower limits | ICC (2,1) | 95%CI upper; lower limits | SEM, ms | %SEM | MDC$_{95}$, ms | %MDC$_{95}$ | r (p) |
|---|---|---|---|---|---|---|---|---|---|---|---|
| Straight | | | | | | | | | | | |
| ON | Rt | 1,236.22 (116.84) | 3.17 (1.97) | −0.72; 7.06 | 0.81 | 0.75; 0.86 | 25.33 | 0.02 | 70.22 | 0.06 | 0.27 (<0.01) |
| | Lt | 1,211.97 (120.67) | −3.97 (0.96) | −5.86; −2.07 | 0.96 | 0.94; 0.98 | 12.90 | 0.01 | 35.76 | 0.03 | 0.02 (0.83) |
| EF | Rt | – | −9.30 (2.17) | −1358; −5.03 | 0.96 | 0.93; 0.97 | 29.17 | 0.02 | 80.85 | 0.07 | 0.36 (<0.01) |
| | Lt | – | −2.33 (1.27) | −4.83; 0.18 | 0.99 | 0.98; 0.99 | 16.35 | 0.01 | 45.31 | 0.04 | 0.07 (0.41) |
| PO | Rt | – | −4.15 (1.94) | −7.99; −0.31 | 0.97 | 0.96; 0.98 | 25.16 | 0.02 | 69.73 | 0.06 | 0.16 (0.04) |
| | Lt | – | −4.89 (2.01) | −8.87; −0.91 | 0.97 | 0.96; 0.98 | 26.17 | 0.02 | 72.54 | 0.06 | 0.23 (<0.01) |
| Curve | | | | | | | | | | | |
| ON | Rt | 885.80(107.42) | 0.92 (1.15) | −1.34; 3.18 | 0.91 | 0.89; 0.93 | 18.13 | 0.02 | 50.26 | 0.06 | 0.14 (0.03) |
| | Lt | 888.24(108.30) | −2.97 (0.74) | −4.42; −1.52 | 0.95 | 0.94; 0.97 | 11.99 | 0.01 | 33.23 | 0.04 | 0.04 (0.49) |
| PO | Rt | – | 26.29 (2.50) | 21.38; 31.21 | 0.89 | 0.72; 0.94 | 47.35 | 0.05 | 131.25 | 0.15 | 0.13 (0.04) |
| | Lt | – | −6.11 (1.64) | −9.34; −2.89 | 0.96 | 0.95; 0.97 | 26.57 | 0.03 | 73.64 | 0.08 | 0.31 (<0.01) |

**Notes.**

Note that stroke duration is presented only for ON since it is the intervals between each stroke

SD, Standard deviation; ON, Onset; EF, Edge-flip; PO, Push-off; Rt, Right side; Lt, Left side; d, Difference between the two measurements; LOA, Limit of Agreement; ICC, Intraclass correlation coefficients; SEM, Standard error of measurement; MDC$_{95}$, Minimal detectable change; %SEM, SEM/stroke; %MDC$_{95}$, MDC$_{95}$/stroke; r (p), correlation coefficient ( *p*-value).

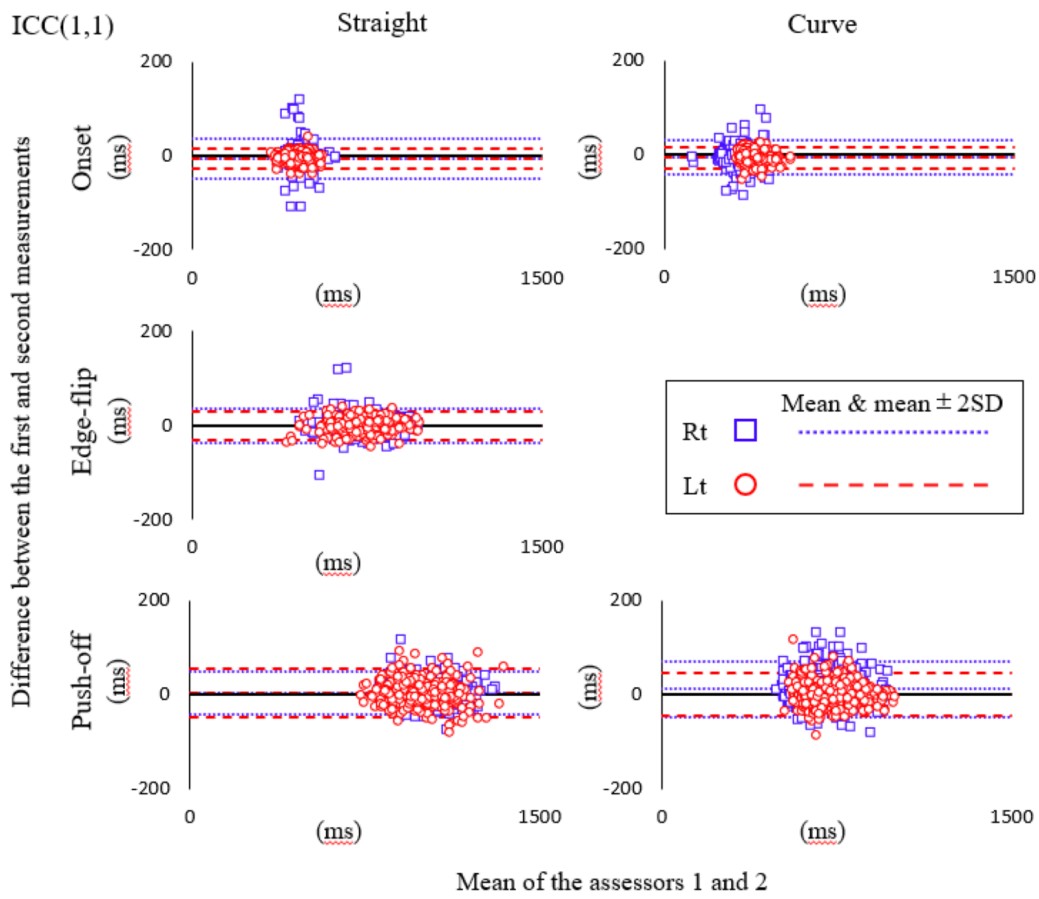

**Figure 5  Bland-Altman plot for intra-rater reliability.** Blue square mark, Right lower limb; Red circle mark, Left lower limb; Fine dotted line, mean and mean ± 2SD of the difference for right; Coarse dotted line, mean and mean ± 2SD of the difference for left.

error was less than $MDC_{95}$. The Bland-Altman plot demonstrated that all items were within mean ± 2SE, and no obvious systematic error was verified by visual observation.

The inter-rater reliability was also high, as demonstrated by ICC(2,1) values of more than 0.8 for all items. Our quantitative analysis showed that multiple items exhibited proportional errors. Additionally, several items showed additive errors with LOAs not crossing zero. However, the directionality of the error was not consistent, and the range of any additive error was less than $MDC_{95}$. Bland-Altman plots showed that all items were within mean ± 2SE and no obvious systematic error was verified by visual observation.

The range of measurement error for both intra- and inter-rater reliability was small, ranging from 0.56 to 26.29 ms. Although some of the items showed additive and systematic errors, both types of errors were less than or equal to $MDC_{95}$. Therefore, both additive and systematic errors observed in this study are considered to be within the range of chance errors and negligible. The precision of the movement phase identification in our methodology is comparable or slightly better than previous studies in skiing (*Meyer et al., 2022*; *Myklebust, Losnegard & Hallén, 2014*), walking (*Romijnders et al., 2021*; *Hansen et al.,*

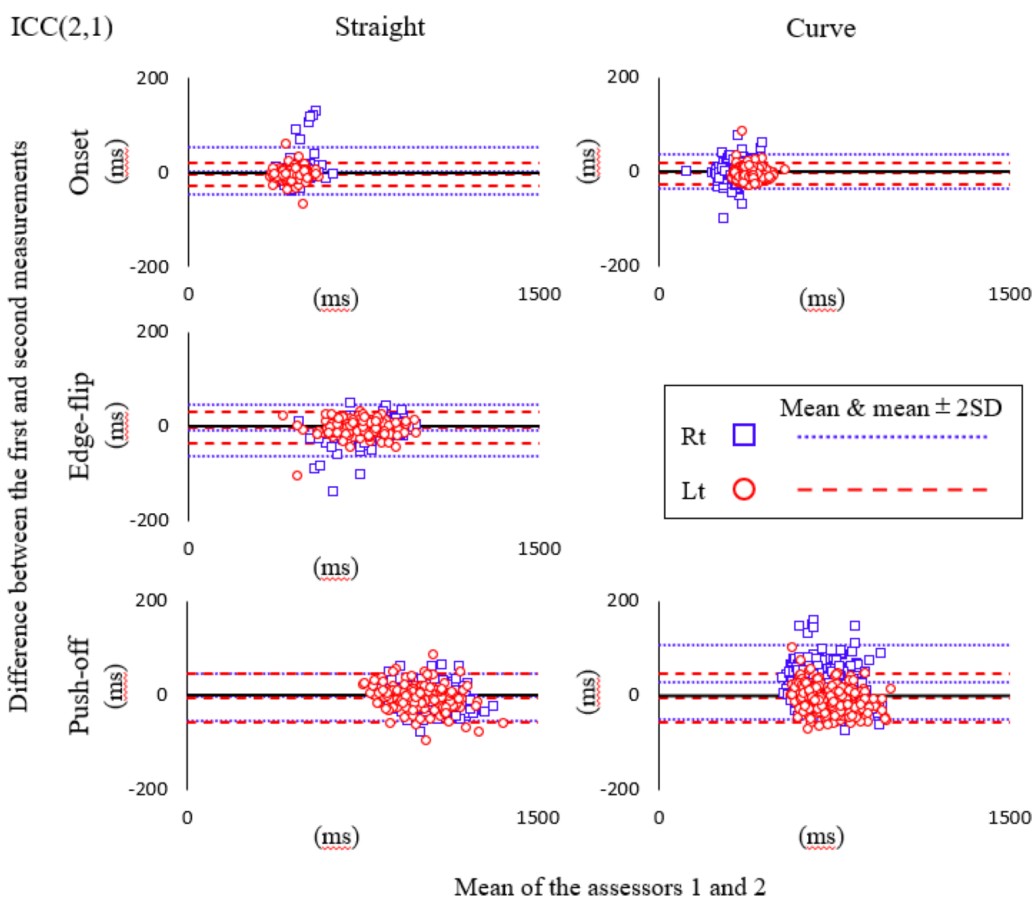

**Figure 6** **Bland-Altman plot for inter-rater reliability.** Blue square mark, Right lower limb; Red circle mark, Left lower limb; Fine dotted line, mean and mean ± 2SD of the difference for right; Coarse dotted line, mean and mean ± 2SD of the difference for left.

*2022*), and running (*Falbriard et al., 2018*). These errors may be partly solved by adopting a measurement system with higher temporal resolution (*Falbriard et al., 2018*).

Furthermore, both intra-rater reliability and inter-rater reliability were relatively lower for the right lower limb during the curve. This may be attributed to the significant difference in the displacement pattern of the knee flexion angle in the right lower limb during the curve compared to the left side and straight. During speed skating, the displacement of the knee flexion angle has two flexion peaks in one stroke, with a tendency for the left lower limb in straights and curves to have a large first peak (Fig. 3). However, the second peak tends to be greater in the right lower limb during curves (Fig. 4). The distinct knee flexion angle displacement profile may have influenced the acceleration profile and angular velocity patterns of the Shank sensor at the ice landing, leading to greater errors in movement event identification. Nonetheless, the high ICC values observed for both intra-rater reliability and inter-rater reliability indicate that the criteria used in this study were reliable and useful for identifying movement events during speed skating. Kinematic analysis of speed skating

can be conducted using precise movement event identification presented in our study to clarify the kinematic characteristics associated with higher performance and injury risk.

This study is subject to several limitations. Firstly, the data were acquired during an experimental race with sensors attached, which might have impacted the skaters' performance. However, the difference between personal best times and experimental race times was less than 1 s, indicating that IMU measurements allow skaters to perform near their best during the data collection. Secondly, our findings are specifically applicable to the 500 m event in young, competitive long-track speed skaters aged 15–22 years. Consequently, the applicability of these results to different skating distances, such as 1,500 m or 10,000 m, and different age categories remains to be established and requires further investigation. Specifically, the acceleration at ON and OFF and knee flexion angle profiles in the long-distance category (*e.g.*, 10,000 m) may differ from those in the short-distance category, requiring further investigation. Secondly, the focus of this study was limited to three key events during the stance phase (onset, edge-flip, and push-off), which is a critical period for velocity generation in skating. However, the validity and reliability of identifying movement phases during the swing phase were not addressed and should be a subject of future research. Furthermore, while the validity of ON and OFF detection using IMUs was established in a previous study (*Tomita et al., 2021*), the validity of EF detection with IMUs needs to be confirmed in future research. This highlights a need for more comprehensive studies to understand movement phase identification across various phases and distances in speed skating.

## CONCLUSIONS

This study examined the reliability of IMUs in identifying events during long-track speed skating runs. The results showed that the kinematic measurement with the IMUs was highly reliable in identifying all events. In addition to the validity of the IMUs for identifying events during long-track speed skating in previous studies (*Tomita et al., 2021*), the present study also showed a high degree of reliability in identifying events during skating. The high reliability of the IMUs in identifying events during long-track speed skating, in addition to its validity in previous studies, suggests that the IMUs may be useful in identifying phases of motion during skating.

## ACKNOWLEDGEMENTS

We thank Takasaki University of Health and Welfare and the skating club of Tsumagoi High School in Gunma Prefecture for their cooperation in this study.

### Funding

Yosuke Tomita was supported by Kakenhi (Grant-in-Aid for Early-Career Scientists No. 19K20011). The funders had no role in study design, data collection and analysis, decision to publish, or preparation of the manuscript.

## Grant Disclosures

The following grant information was disclosed by the authors:
Kakenhi: 19K20011.

## Competing Interests

The authors declare there are no competing interests.

## Author Contributions

- Tomoki Iizuka conceived and designed the experiments, performed the experiments, analyzed the data, prepared figures and/or tables, authored or reviewed drafts of the article, and approved the final draft.
- Yosuke Tomita conceived and designed the experiments, performed the experiments, analyzed the data, prepared figures and/or tables, authored or reviewed drafts of the article, and approved the final draft.

## Human Ethics

The following information was supplied relating to ethical approvals (*i.e.*, approving body and any reference numbers):

The study was approved by the Ethics Committee of Takasaki University of Health and Welfare (No. 1904).

## Data Availability

The raw data are available in the Supplemental File.

## Supplemental Information

Supplemental information for this article can be found online at http://dx.doi.org/10.7717/peerj.18102#supplemental-information.

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
