# Peer review of "Reliability of motion phase identification for long-track speed skating using inertial measurement units"

_PeerJ, doi:10.7717/peerj.18102_

## Round 0.1 · original submission · Major Revisions

Overall, the reviewers are positive regarding the paper, but have some serious concerns that should be addressed. Please make also sure the paper is written in appropriate English. Maybe a native speaker would be helpful.

Reviewer 1 ·

Basic reporting

1.The figures are unclear.
2.The topic lacks novelty and significance.
3.Validating the reliability of IMUs for motion identification does not enhance speed skating performance.

Experimental design

no comment

Validity of the findings

no comment

Additional comments

no comment

Reviewer 2 ·

Basic reporting

Clear and Professional English
- Although generally clear, there are some sentences that could be simplified or improved in structure to enhance readability. Consider additional proofreading by a native speaker or professional editor.
Literature References and Field Context
- Adding more details about previous studies and how this research complements or differs from them would provide a stronger context. Specific examples from previous studies showing the gap filled by this research would be very useful.
Article Structure, Figures, Tables, and Raw Data
- The structure of the article adheres to professional standards. There is a clear division between the introduction, methods, results, and discussion sections. Figures and tables are presented clearly and informatively.
Self-contained with Relevant Results
- The article is self-contained and includes all necessary information to understand the research results. The results presented are relevant to the hypotheses proposed.

Experimental design

Original Primary Research
- Ensure to explicitly mention how this research expands or adds new knowledge compared to previous studies. This will help readers understand the unique contribution of this research.
Well-defined Research Question
- Although the research question is clear, adding more details about the theoretical background and why this question is important in a broader context can enhance understanding and relevance.
Rigorous Investigation with High Technical and Ethical Standards
- The research is conducted with high technical and ethical standards. The data collection process using IMU is described in detail, including sensor placement and measured parameters.
Methods Sufficiently Detailed for Replication
- The methods used are described in sufficient detail to allow replication. The explanation includes experimental design, tools used, and data collection procedures

Validity of the findings

Meaningful Replication Encouraged
- The article does not directly assess impact and novelty but encourages meaningful replication by stating the rationale and benefits to the literature.
Robust Underlying Data Provided
- All underlying data have been provided; they are robust, statistically sound, and controlled. The statistical analysis used is appropriate and supports the conclusions drawn.
Well-stated Conclusions
- The conclusions are well stated, linked to the original research question, and limited to the supporting results.

Additional comments

To improve its quality, we recommend additional proofreading by a native English speaker or professional editor to enhance readability. Strengthen the context by adding more details about how your research complements or differs from previous studies, and consider including more visualizations for key data. Clearly state how your research expands on existing knowledge, and provide additional background on the theoretical importance of your research question. Ensure that your methodology is detailed enough for replication and discuss the potential implications of your findings in real-world applications. These revisions will significantly enhance the quality and impact of your manuscript.

·

Basic reporting

The manuscript is very well presented.

Experimental design

Within the journal's scope and research question well defined. Methods well presented and clearly to a high standard.

Validity of the findings

Data provided and conclusions well stated.

Additional comments

Thank you for the opportunity to review "Enhancing speed skating performance: A comprehensive analysis of IMU-based motion phase identification reliability". I send our congratulations to the authors to conduct this study. The validation performed in this study is interesting and may lead to the proposal of effective measures for future analyzers involved in speed skating. However, some modifications are required to better understand and communicate the results to the reader.


*The following information needs to be corrected.
Introduction
・L54
“out-edge" and "in-edge".
Please add an explanation of this term so that people who are unfamiliar with speed skating can understand it.

Materials & Methods
・L107-108, Figure 1
In the text, the terms "knee flexion angle" and "knee joint flexion angles" are used, while the caption of Figure 1 states "knee joint angle" and within Figure 1 "knee angle". L240 also uses the notation "knee-joint angle". If the meaning of each of these terms is the same, please ensure that the terms are consistent in other places. Please be careful to be consistent in the use of other words and phrases in the paper as well and check the entire text again.

・Figures 1, 2, and 3
For all graphical data, please provide the title, units, scale, and zero position of the vertical and horizontal axes.

Results
Tables 3 and 4
・Please explain "Rt" and "Lt" listed in the table.
・Please provide a definition of “Stroke duration” in the Materials & Methods section.
・Why is the stroke duration for curve skating not listed?
・What do the numbers in parentheses in the Stroke duration and d entries represent?

References
・Some papers list the authors using "et al.," while others list all authors. I think all authors should be listed without using "et al.,". Please read the submission rules carefully and make the correct entries. Also, all papers should be listed in a uniform manner.
・5 : Yuda was the author of this study. Please check again with great care to ensure that other information, such as author information, title, and journal name, is listed correctly.
Yuda Jun et al. 2007. doi: 10.1123/jab.23.2.128


*The remarks listed below should be addressed if the author feels they are necessary. They do not necessarily need to be corrected.

Introduction
Much of what is discussed in the introduction of this paper overlaps with the introduction content of a previous study by the authors' group (Tomita et al., 2021). Introduction content would need to be improved to explain the original purpose and importance of this study. To this end, I suggest that the authors explain in more detail what kind of validation was conducted in their previous study (Tomita et al., 2021), what was clarified and what was not, etc., and clearly explain how it relates to this study and how they arrived at their objective.

・L51-52
“The stance phase, crucial for velocity generation…"
On what basis do you state that the stance phase is crucial for generating velocity?
If possible, we encourage you to explain this by citing prior research.

・L53
“Turn Back."
Is this terminology commonly used in the field of competition or in previous studies?

・L56-57
It is recommended to cite several previous studies that use multiple video cameras to capture skaters and the three-dimensional panning direct linear transformation (3D panning DLT) technique to calculate and analyze the coordinates.

・L66-67
There is a previous study (van der Kruk E. et al., 2018) that analyzed speed skating using IMU, but it is not cited here, which seems unnatural. The reader may wonder if the authors intentionally did not cite this paper to claim novelty. If the author feels it is necessary, please consider adding it to the citation.
van der Kruk E, Schwab AL, van der Helm FCT, Veeger HEJ. Getting in shape: Reconstructing three-dimensional long-track speed skating kinematics by comparing several body pose reconstruction techniques. J Biomech, 69: 103-112, 2018. doi: 10.1016/j.jbiomech.2018.01.002

・L75-77
I suggest that the authors explain in more detail what kind of validation was done in their previous study (Tomita et al., 2021), what was clarified and what was not clarified, etc., and explain in more detail the relationship to this study and how they arrived at their objectives. It would make the position of this study clearer and easier to understand.

Materials & Methods
・Table 1
It is recommended that age and personal best time information be broken down by gender and that mean and standard deviation values be given.

・Table 2
I suggest using photos or illustrations of skaters wearing IMUs to illustrate.

・L105-106
Did the skaters perform their skates on single-track or double-track? As Tokachi Oval is a C track type, the length of the straight section is 110.43m, the length of the inner lane curve is 83.25m, and the length of the outer lane curve is 95.82m. If exact information is not available, it is recommended to state "approximately 100m".

・L106
It is recommended that you define the term "stroke" to refer to the event to which event. Many of the papers listed below define "stroke" as the period from blade off to opposite blade off. They also define “one cycle” as a series of left and right strokes. Following these definitions, I think that the "cycle" is close to the section from the blade off to the next ipsilateral blade off, which is defined as a "stroke" in this paper. However, if the author thinks it is appropriate to use "stroke," then there is no problem. In this case, I recommend stating that one stroke is defined as the interval from the blade off to the next ipsilateral blade off.

・Figure 1
I suggest placing a series of photos or stick pictures of skaters skating along the horizontal axis of the graph.

・Figure 2
I think that the figure shown here is the data obtained during straight skating. To avoid confusion, I recommend that the data for the left lower limb of curve skating be shown in a separate figure.

・L119-120
Please explain "shank anterior-posterior (AP) acceleration." In addition, where on the graph are the data for shank anterior-posterior (AP) acceleration shown? If not, please consider this issue.

・L126-127
I suggest that the angle definitions of "knee flexion angle" and "roll angle of the Shank sensor" are shown in the figure. Please indicate the magnitude and positive/negative direction of each in the figure or text.

・L135-137
Since there is no "TB" for curve skating, it is recommended that this description be divided into two sentences, one for straight skating and one for curve skating.

Results
・L178
If possible, please provide the 500m time for the experimental race.

Discussion
・L246-255
Are not there any other limitations to this study?
For example, during an experimental race, were skaters able to skate at their best while wearing the IMU device? Is the data obtained from this experiment the true performance of a skater?
Is it possible to perform the same detection in the first 100 m of the 500 m?
The validity of onset-offset detection has already been verified in the authors' previous study (Tomita et al., 2021), has the validity of turn-back detection verified?

·

Basic reporting

The manuscript would benefit from further refinement and enhancement of its English language usage to meet academic standards. e.g.

Readers should not have to guess the meaning of acronyms. For instance, the first mention of "ICC" should be accompanied by its full name, "Intraclass Correlation Coefficient."

The manuscript should clearly define and explain technical terms, especially those that are not commonly used in the field of speed skating. A visual aid, such as a diagram or illustration, could be included to clarify the "stance phase" and "swing phase."

Experimental design

Table 2 provides a general description of the sensor placement but lacks precision, missing coordinates. It is suggested to include a diagram for more precise localization of the sensors.

The definitions and calculations for ON, TB, and PO are described only in narrative form, lacking quantitative formulas, which compromises the rigor of the study. Moreover, considering the high variability of sensor test data, the criteria described should likely be applied to filtered test data. It is recommended to clarify this point.

Validity of the findings

Figures 2 and 3 are missing horizontal axes; furthermore, upon comparison, Figure 3 lacks data for the Shank Roll.

In speed skating, the technical characteristics of the two legs differ during various phases of skating during straight or curve segments. This paper seems to lack explanation and analysis of these differences, which may affect the calculations of ON, TB, and PO.

The first limitation mentioned at L246 does not seem reasonable. Regardless of the distance, the basic movements and patterns in speed skating are consistent, with differences only in stride length and frequency. Therefore, the methods presented in this paper should also be applicable to longer distances.

Additional comments

The manuscript presents data from correlation analysis, reaffirming the applicability of IMUs; however, it lacks a thorough analysis of the implications for sports, specifically speed skating. The paper is deficient in describing the practical applications and reference value of the data obtained for the sport of speed skating.

---

## Round 0.2 · accepted · Accept

I am pleased to inform you that the reviewers and I agree with the edits. The manuscript is now ready for publication

·

Basic reporting

The manuscript is very well presented.

Experimental design

Within the journal's scope and research question well defined. Methods well presented and clearly to a high standard.

Validity of the findings

Data provided and conclusions well stated.

Additional comments

L110, L119
It is unnatural that the total does not amount to 400 m.
If the straight section is described as “approximately 110 m,” it is recommended that the curve section be described as “approximately 90 m.

There is no other content that needs to be corrected. It has been corrected very carefully.

·

Basic reporting

no comment

Experimental design

no comment

Validity of the findings

no comment

Additional comments

no comment